# Bryophytes are predicted to lag behind future climate change despite their high dispersal capacities

F. Zanatta[1,10], R. Engler[2,10], F. Collart [1,10], O. Broennimann[3,4], R. G. Mateo [5,6], B. Papp[7], J. Muñoz [8], D. Baurain [9], A. Guisan[3,4,11] & A. Vanderpoorten[1,11✉]

The extent to which species can balance out the loss of suitable habitats due to climate warming by shifting their ranges is an area of controversy. Here, we assess whether highly efficient wind-dispersed organisms like bryophytes can keep-up with projected shifts in their areas of suitable climate. Using a hybrid statistical-mechanistic approach accounting for spatial and temporal variations in both climatic and wind conditions, we simulate future migrations across Europe for 40 bryophyte species until 2050. The median ratios between predicted range loss vs expansion by 2050 across species and climate change scenarios range from 1.6 to 3.3 when only shifts in climatic suitability were considered, but increase to 34.7–96.8 when species dispersal abilities are added to our models. This highlights the importance of accounting for dispersal restrictions when projecting future distribution ranges and suggests that even highly dispersive organisms like bryophytes are not equipped to fully track the rates of ongoing climate change in the course of the next decades.

[1]Institute of Botany, University of Liège, B-4000 Liège, Belgium. [2]Vital-IT, SIB Swiss Institute of Bioinformatics, CH-1015 Lausanne, Switzerland. [3]Department of Ecology and Evolution (DEE), University of Lausanne, CH-1015 Lausanne, Switzerland. [4]Institute of Earth Surface Dynamics (IDYST), University of Lausanne, CH-1015 Lausanne, Switzerland. [5]Department of Biology (Botánica), Universidad Autónoma de Madrid, E-28049 Madrid, Spain. [6]Centro de Investigación en Biodiversidad y Cambio Global (CIBC-UAM), Universidad Autónoma de Madrid, E-28049 Madrid, Spain. [7]Department of Botany, Hungarian Natural History Museum, H-1431 Budapest, Hungary. [8]Real Jardín Botánico (CSIC), E-28014 Madrid, Spain. [9]InBioS-PhytoSYSTEMS, Eukaryotic Phylogenomics, University of Liège, B-4000 Liège, Belgium. [10]These authors contributed equally: F. Zanatta, R. Engler, F. Collart. [11]These authors jointly supervised this work: A. Guisan, A. Vanderpoorten. ✉email: a.vanderpoorten@uliege.be

Despite a growing number of climate change mitigation policies, anthropogenic greenhouse gas emissions have continued to increase since the pre-industrial era. Globally, an average warming of 1.0 °C as compared to pre-industrial levels has been reported and is expected to reach 1.5 °C between 2030 and 2052, with substantial regional variations. In the Arctic for instance, two to three times higher warming rates than the global annual average are expected[1]. The impacts of this global warming on biodiversity have been largely documented[2] and climate change has been identified as one of the major biodiversity threats[3,4], with the worst-case scenarios leading to extinction rates that would qualify as the sixth mass extinction in the Earth history[5].

While climate change is making some current habitats unsuitable, it is also expected to create newly suitable areas for species to occupy. The extent to which species have the ability to balance the loss of suitable habitats by shifting their ranges and track areas of suitable climate has, however, been debated[6–8]. Despite reports that many species lag behind climate change[9], nearly as many studies of observed latitudinal changes fall above as below the observed[10]. For plants in particular, empirical evidence for lagged migration is far from clear-cut[11]. While the coincident increase of species richness with climate warming towards high elevations is suggestive of a rapid response of communities to climate change[12], considerable lags in the future response to climate warming have been predicted for Alpine plants[13]. Such lag has also been observed in the field: Rumpf et al.[14] recently reported that 38% of plant species they investigated were not able to colonize all the sites that became climatically suitable to them.

Assessing range shifts and extinction risks involves an assessment of (i) the change in climatically suitable habitats over time and (ii) the species ability to adapt or migrate to track areas of newly suitable climate[15]. In this context, spatially explicit climatic suitability and distribution models (also called species distribution models, or ecological niche models) have been the most widely used tool to assess the impact of projected climate change

on future species distributions and biodiversity patterns[16]. Contrasting model predictions with actual distribution data revealed, however, that a substantial fraction of species are missing from areas projected as suitable[17,18]. This, together with the significant effect of geographic distance on the taxonomic and phylogenetic turn-over of species communities[17,19,20], points to the need to account for dispersal limitations when predicting species distributions under climate change[21,22]. Mounting evidence therefore suggests that approaches integrating mechanical dispersal processes into climatic suitability and distribution models have higher predictive accuracy in forecasting species range shifts than structurally simpler models that only account for species' correlates with climate[23,24].

The primary goal of the present study is to determine the extent to which highly efficient dispersers like bryophytes can mitigate the loss of suitable habitats through rapid colonization of newly suitable areas. The relevance of bryophytes, which represent the second most diversified group of land plants after the angiosperms[25], in range shift studies, is twofold. First, bryophytes hold exceptional importance in the control of global carbon fluxes and climate because of the vast stores of carbon bound-up in peat[26]. In particular, more carbon is stored in *Sphagnum* than in any other genus of plant[27]. Second, bryophytes lack roots and therefore cannot uptake water directly from the water table, making them reliant on atmospheric precipitations. Furthermore, bryophyte species of temperate biomes exhibit lower optima and tolerance to warm temperatures than their angiosperm counterparts[28] (but see ref. [29]). These specific ecophysiological features make bryophytes ideal indicators of the impact of climate change on biodiversity patterns.

Here, we implement a hybrid statistical-mechanistic approach that accounts for temporal and spatial variation of both climatic conditions and wind connectivity to predict potential shifts in distribution across Europe for 40 bryophyte species until 2050, at a spatial resolution of 1 km$^2$. We show that projected rates of range loss largely exceed the proportion of newly suitable habitats

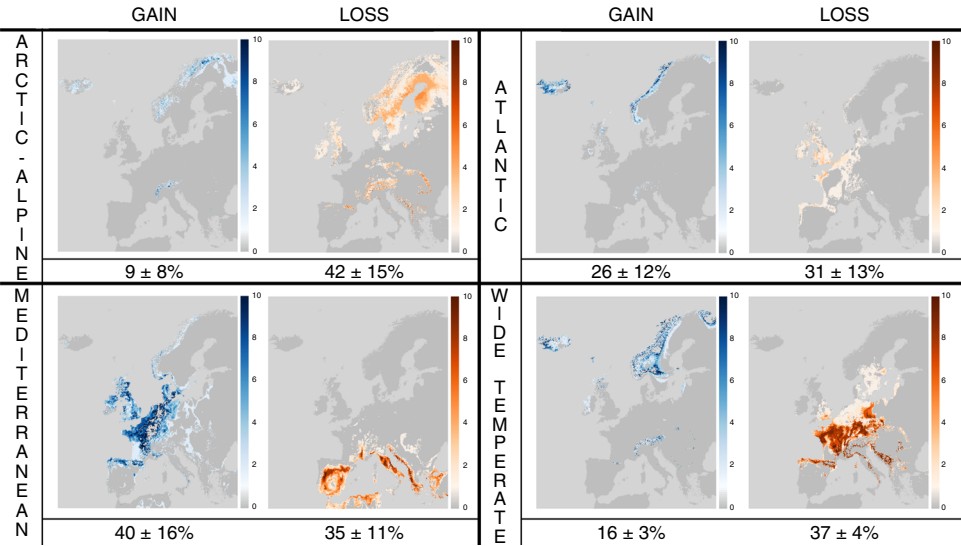

**Fig. 1 Predicted impact of future climate change for the potential distribution of European bryophytes.** The maps represent the distribution of 1 km$^2$ pixels predicted to become climatically suitable and unsuitable in 2050 for 10 representative bryophyte species of each of the four main biogeographic elements in Europe (Mediterranean, Atlantic, wide-temperate and Arctic-Alpine) using ensemble of climatic suitability models with the MPI-ESM-LR Global Circulation Model under scenario RCP8.5 (see Supplementary Fig. 1 for scenario RCP4.5 and Supplementary Figs. 2 and 3 for the two RCPs with the HadGem2-ES Global Circulation Model). Colours represent the proportion of species, computed over 10 species per biogeographical elements (individual maps are available from Figshare, DOI: 10.6084/m9.figshare.8289698), for which a pixel becomes suitable (blue) and unsuitable (red). Numbers indicate the average (±S.D.) percentage of the predicted increase (number of pixels that become suitable in 2050) and loss (number of pixels that become unsuitable in 2050), respectively, of suitable area in 2050 as compared to the extent number of suitable pixels.

that could effectively be colonized, suggesting that even highly dispersive organisms such as bryophytes might not be equipped to track the rates of ongoing climate change in the course of the next decades.

## Results

We predicted range shifts under changing climate conditions until 2050 in 40 bryophyte species representative of the Mediterranean, Atlantic, wide-temperate and Arctic-Alpine biogeographic elements. The climatic suitability models exhibited high average True Skill Statistics (TSS) and Area Under The Curve (AUC) of a ROC plot (Receiver Operating Characteristics) statistics[30] of $0.78 \pm 0.12$ and $0.93 \pm 0.05$ respectively, when models were evaluated against test sets corresponding to 20% of the data (cross-validation). These models did not show any apparent signature of overfitting, as only a very slight increase in AUC and TSS ($0.81 \pm 0.13$ and $0.94 \pm 0.05$, respectively) was observed when these statistics were computed at the level of the entire dataset (Supplementary Table 2).

With the MPI-ESM-LR Global Circulation Model (GCM), the highest relative rates of range loss are predicted for the Arctic-Alpine element, with an average loss of $40 \pm 12\%$ and $42 \pm 14\%$ and an average gain of $9 \pm 7\%$ and $9 \pm 8\%$ under the Representative Concentration Pathway (RCP) 4.5 and 8.5 climate change scenarios, respectively (Fig. 1, Supplementary Table 2 and Supplementary Fig. 1). The highest rates of relative range expansion are predicted for the Mediterranean element, with a $32 \pm 10\%$ ($35 \pm 10\%$) loss against a $38 \pm 14\%$ ($39 \pm 15\%$) gain, due to the clear tendency for a northern shift of the climatically suitable area (Fig. 1 and Supplementary Fig. 1). Similar, but even more dramatic predictions in terms of range loss, with a maximum of $73 \pm 6\%$ in the wide-temperate element, were obtained with the HadGem2-ES GCM (Supplementary Table 2 and Supplementary Figs. 2 and 3).

Simulated colonization rates (i.e., the ratio between the number of effective colonization events and the number of pixels becoming suitable by 2050) are displayed in Fig. 2 and Supplementary Fig. 4 (MPI-ESM-LR GCM under RCP scenarios 8.5 and 4.5, respectively) and Supplementary Figs. 5–6 (HadGem2-ES GCM under RCP scenarios 4.5 and 8.5, respectively).

There was a clear impact of release height on colonization rates, whose median ranged from 4% at 0.03 m to 63% at 10 m for the largest spores at maximum wind speed, and between 59% and 84% at 0.03 m for small and medium-sized spores, respectively, whatever the long-distance dispersal probability. At release heights of 1 and 10 m, colonization rates reached 98% and 99% for small and medium-sized spores, respectively (see Fig. 2 for scenario RCP8.5, with similar trends for scenario 4.5 in Supplementary Fig. 4 and for the HadGem2-ES GCM under RCP scenarios 4.5 and 8.5, Supplementary Figs. 5 and 6). Wind speed mostly played a role for the largest spores, whose colonization rates were 1–57 times higher when maximum vs average wind layers were employed, but its impact was lower for smaller spores. Finally, spore size also substantially impacted colonization rates, with a median <1 to 7% for large spores, 25 to almost 100% for medium-sized spores, and 61 to almost 100% for small spores depending on release height and long-distance dispersal probability under average wind conditions.

Running the simulations beyond 2050 to determine the time-lag of the colonization of newly suitable habitats, i.e., how many years would be needed for species to fully colonize all the climatically suitable habitats after 2050, we found, using release height values based on habitat preferences and maximum wind speed layers, that, on average, 80–100% and 25–70% of the species would need more than 500 years to successfully colonize all the newly suitable habitats when the long-distance dispersal probability was set to 0 and 0.1, respectively (Table 1 and Supplementary Table 1). Depending on climate change scenarios,

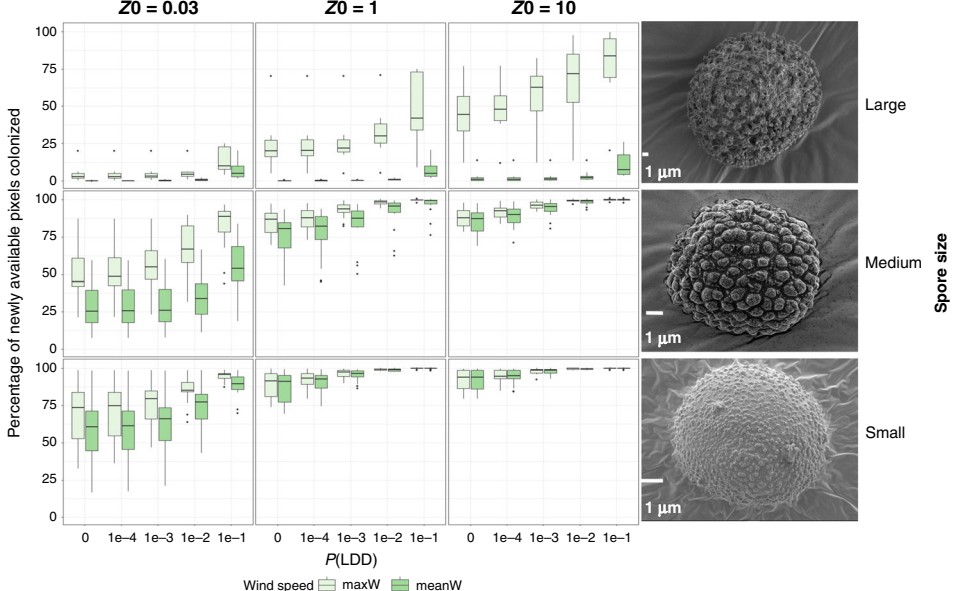

**Fig. 2 Colonization rates of areas predicted to become climatically suitable due to climate change in European bryophytes.** The box-plots (showing the 1st and 3rd quartiles (upper and lower bounds), 2nd quartile (centre), 1.5* interquartile range (whiskers) and minima-maxima beyond the whiskers) represent simulated colonization rates expressed as the ratio (*100), averaged over 30 replicates, between the number of effective colonization events (including effective colonization events that eventually got extinct at the end of the simulation) and the total number of pixels becoming suitable by 2050 in 40 selected bryophyte species in Europe as a function of spore size (a: <20 μm; b: 20–50 μm; c: >50 μm), release height Z0, wind speed, and probability of long-distance dispersal P(LDD), with the MPI-ESM-LR Global Circulation Model under climate change scenario RCP8.5 (see Supplementary Fig. 4 for scenario 4.5). The right panel illustrates selected SEM photographs of spores of *Scleropodium touretii* (small spores), *Ulota bruchii* (medium-sized spores) and *Archidium alternifolium* (large spores).

**Table 1 Time-lag of the colonization of newly suitable habitats in 2050 for 40 selected bryophyte species in Europe, as assessed by MigClim dispersal simulations under changing climate conditions defined by the MPI-ESM-LR (MPI) and HadGem2-ES (HE) Global Circulation Models and climate change scenarios RCP4.5 and 8.5.**

| MPI4.5 | MPI8.5 | HE4.5 | HE8.5 |
|---|---|---|---|
| LDD = 0 | | | |
| 80% | 98% | 100% | 98% |
| 10% | 0% | 0% | 2% |
| 111 ± 126 years | 149 years | – | – |
| LDD = 0.1 | | | |
| 70% | 35% | 25% | 27% |
| 22% | 25% | 25% | 30% |
| 21 ± 6 years | 66 ± 85 years | 86 ± 119 years | 98 ± 141 years |

LDD = 0 and 0.1 refer to the probability of long-distance dispersal implemented by the model, respectively. For each GCM, climate change scenario and LDD probability, we indicate (i) the percentage of species that failed to colonize all newly suitable habitats by 2050 after 500 years (top); (ii) the percentage of species that fully colonized all newly suitable habitats by 2050 and are hence at equilibrium with climate conditions (middle); and (iii) for the remaining species, the average (±SD) number of years required to fully colonize all newly suitable habitats after 2050 (bottom) (see Supplementary Table 2 for detailed information for each species).

only 0–10% and 22–30% of the species fully colonized all newly climatically suitable areas by 2050 when the long-distance dispersal probability was set to 0 and 0.1, respectively, and were hence at equilibrium with the environment (Table 1). The remaining species required, on average, 21 ± 6 years to 98 ± 141 years after 2050 before the colonization rate reached 100% of the newly suitable pixels depending on climate change scenarios and LDD probability (Table 1).

The ratio between the rates of range loss and gain at the end of the simulation in 2050 is displayed in Fig. 3, and evidences a clear pattern of substantial range contraction. Median ratios between predicted range loss vs expansion until 2050 across species ranged between 1.6 and 3.3 depending on climate change scenarios when only shifts in climatic suitability were considered, but between 34.7 and 96.8 depending on climate change scenarios and dispersal kernels when effective colonization was considered (Supplementary Table 2). With the global circulation model HadGem2-ES, the median loss/gain ratio was the highest in the case of the wide-temperate element (~75:1) as compared to a median ratio of slightly more than 50:1 for the other elements. With the MPI-ESM-LR Global Circulation Model, the median loss/gain ratio was the highest in the case of the Alpine-Artic element (~50:1) as compared to a median ratio of slightly more than 25:1 for the other elements.

## Discussion

Simulating wind dispersal across a variable landscape is a challenging task because spatial variations in wind speed, topography and canopy structure affect the probabilities of colonization during the transportation and deposition phases[31]. Substantial variation in environmental heterogeneity affecting both climatic suitability and the ability of species to disperse therefore required developing a spatially explicit modeling framework. Previous studies that attempted at simulating wind dispersal under changing environmental conditions either (i) implemented constant dispersal kernels or randomly sampled prior distributions of migration rates at large scales[11,32,33], or (ii) used detailed models based on local wind conditions and accounting for population dynamics, but could only do so over a limited geographical extent[34,35]. In contrast, our approach allowed us to assess the impact of climate change on a group of wind-dispersed plants, bryophytes, by taking into account local variations in niche suitability and dispersal limitations at a continental scale.

Our simulations are, however, based on a number of simplifications due to limitations in the availability of empirical data. These limitations include, most importantly, the assumptions that dispersal is isotropic, that newly colonized cells are readily considered as sources, thereby ignoring demography, that there is no competition and that microclimatically suitable pixels can serve as migration sources. These assumptions result in an over- rather than an under-estimation of colonization rates, so that our approach is conservative in the sense that, as in Dullinger et al.[11], our results should be at the upper bound of those achievable. Despite this, only a portion of the areas projected to become climatically suitable are predicted to be effectively colonized by 2050. Median ratios between predicted range loss and expansion across species ranged between 1.6 and 3.3 depending on climate change scenarios when only shifts in climatic suitability were considered and increased to between 34.6 and 96.8 depending on dispersal kernels when effective colonization was considered. There was, however, substantial regional variation, as the Arctic-Alpine species pool was predicted to experience the highest range loss (39 ± 15%), whereas the wide-temperate species pool exhibited the lowest net decrease of 18 ± 4%, followed by the Mediterranean species pool with 24 ± 14%. While the Arctic-Alpine species pool was indeed identified as one of the most sensitive to climate warming, the results reported here for Mediterranean bryophytes sharply contrast with the alarming predicted range loss of 60% reported in angiosperms[36]. We suggest that this difference is due to the much wider distribution range, higher dispersal capacities and, potentially, broader climatic niche of Mediterranean bryophytes as compared to their angiosperm counterpart. This is best illustrated by the large differences in rates of local endemism between the two groups, as more than 60% of Mediterranean endemic angiosperm species are restricted to a single region[37] and are, hence, prone to extinction if they fail to colonize newly suitable areas, whereas there is no local endemism reported to date in the Mediterranean bryophyte flora[38].

While bryophytes successfully back-colonized areas of suitable climate since the end of the last glacial maximum, 18,000 years ago[39], our results suggest that, at best, ~30% of the species would be at equilibrium with their environment by 2050. This indicates that bryophytes are not equipped to track the very fast rates of ongoing climate change projected for the course of the next decades. Although recent evidence for synchronized increases in species richness towards high elevations and global warming points to rapid colonization potential of newly available habitats[12], our results, together with other analyses investigating species-specific responses[13,14,40], suggest that changes in diversity patterns tend to mask considerably the delays observed at the level of individual species. In fact, a growing body of evidence supports the idea that plant species spread rates are consistently expected to be much lower than the velocity of climate change[11,33–35]. This highlights the crucial role of integrating dispersal when attempting to predict future distribution ranges[22–24], even in apparently highly dispersive organisms like bryophytes.

## Methods

The methodological framework for simulating the dispersal of bryophytes under changing climate conditions is presented in Fig. 4. A grid of pixel-specific environmental conditions and dispersal kernels, combining information on species dispersal traits, local wind conditions, as well as landscape features affecting dispersal by wind, is generated and used as input in simulations of species dispersal in the landscape under changing climate conditions.

**Data sampling.** The European bryophyte flora includes 1817 native or naturalized species[41]. Because information on bryophyte species distribution is scarce and very heterogeneous, challenging the application of climatic suitability models[42], we

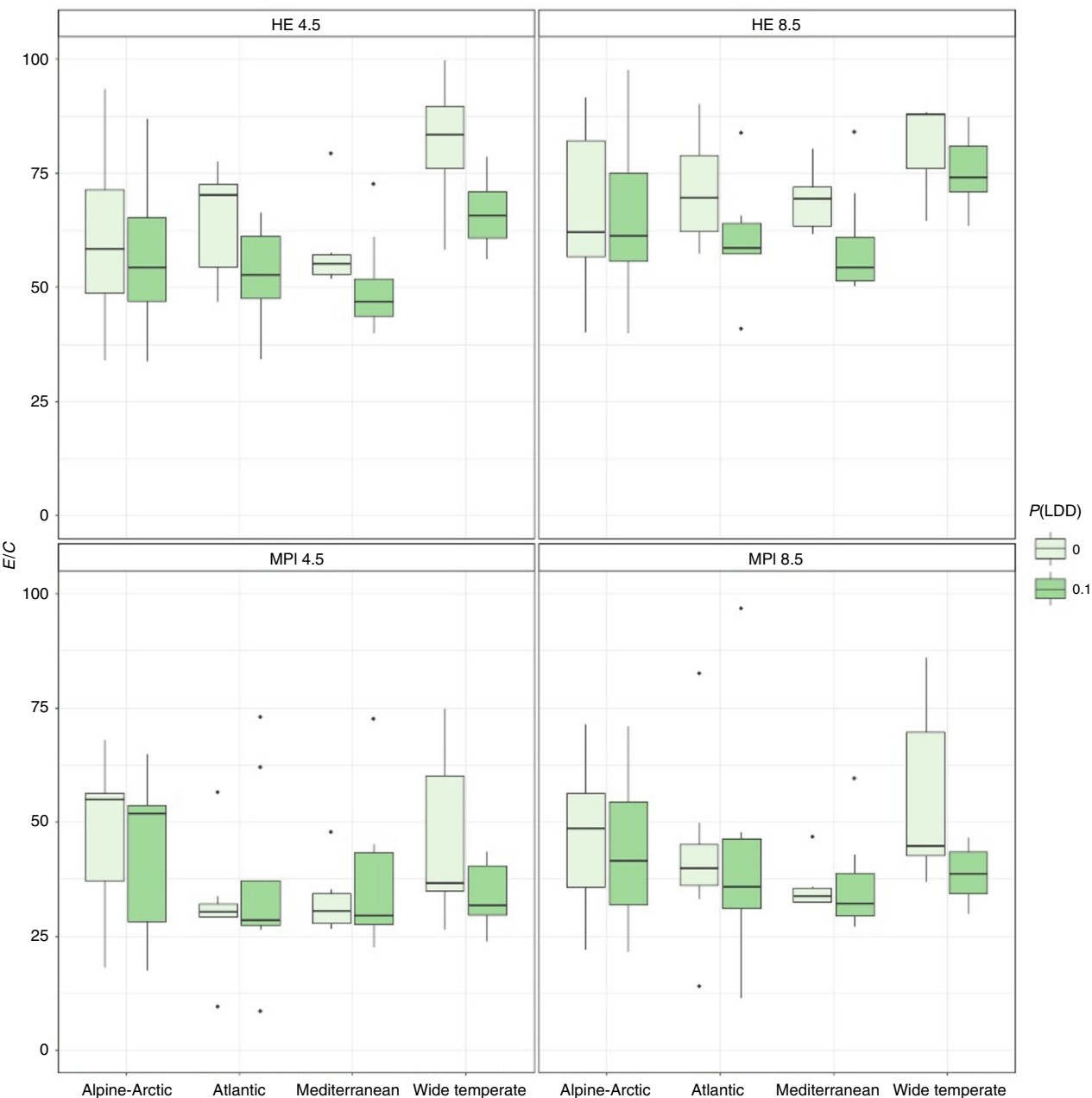

**Fig. 3 Predicted rates of future extinction and colonization of areas becoming newly suitable due to climate change in European bryophytes.** The box-plots (showing the 1st and 3rd quartiles (upper and lower bounds), 2nd quartile (centre), 1.5* interquartile range (whiskers) and minima-maxima beyond the whiskers) represent the ratio $E/C$, averaged over 30 MigClim replicates, between the predicted rate of range loss $E$ (fraction of initially suitable cells that became unsuitable by 2050) and the rate of simulated effective colonization events $C$ (fraction of newly suitable cells by 2050 that were effectively colonized) in 40 selected bryophyte species from the four main biogeographic elements in Europe. Results are shown for two selected dispersal kernels (release height set depending on species habitat preferences, maximum wind layer, and long-distance dispersal probability of 0 and 0.1), two global circulation models (MPI: MPI-ESM-LR and HE: HadGem2-ES) and two climatic scenarios (RCP4.5 and 8.5).

selected 10 species based upon their representativeness for each of the four main biogeographic elements (i.e., groups of species sharing similar distribution patterns), namely the Arctic-Alpine, Atlantic, Mediterranean, and wide-temperate elements (Supplementary Table 2). For each of these species, we downloaded data from the Global Biodiversity Information Facility (https://www.gbif.org). We excluded data collected before 1960, which represented, on average, 41 ± 12% of the data available, for two reasons. First, old records often lack sufficiently precise location information. Second, we wanted to avoid a potential mismatch between old observations and current climate conditions used for modeling. To complete these data and generate a dataset across the entire range of each species in Europe, we specifically performed a thorough literature review to document their occurrence from more than 600 sources. Only points that were separated by at least 0.1° from each other were subsequently retained for modeling ("ecospat.occ.

desaggregation" function in Ecospat 3.1[43]) to avoid sampling bias and reduce the risk of spatial autocorrelation. Altogether, the number of observations available for each species ranged between 55 and 34,035 (database available from Figshare, https://doi.org/10.6084/m9.figshare.8289650).

Average spore diameter was recorded for each species from Zanatta et al.[44] and references therein. Species unknown to produce sporophytes were assigned a spore size of 150 μm to take dispersal through larger asexual propagules into account. Spore settling velocities $V_t$ and release height (0.03, 1 and 10 m, which roughly correspond to habitat preferences for ground-dwelling, saxicolous, and epiphytic species, respectively) were determined for each species (Supplementary Table 2) following Zanatta et al.[44].

Nineteen bioclimatic variables, averaged over the period from 1970 to 2000, were retrieved from WorldClim 1.4 at a resolution of 30 arc-seconds[45]. Although

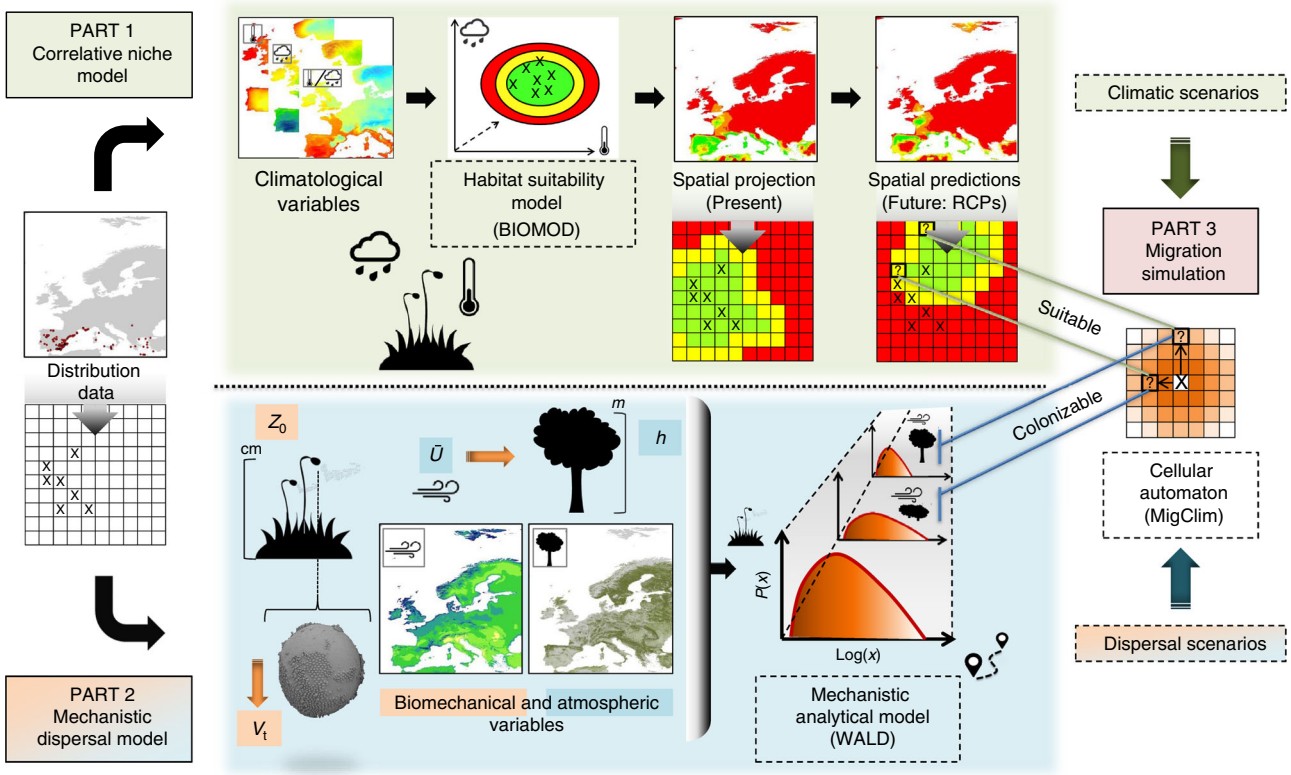

**Fig. 4 Overview of workflow implemented in the present study to integrate mechanistic dispersal kernels and correlative climatic suitability models in simulations of future wind-dispersed species distributions under climate change.** Species distribution data (left) are combined with climatic variables to produce climatic suitability models that are calibrated under present and projected under future climatic conditions (Part 1) and used to build mechanistic dispersal models (Part 2). The latter combine species intrinsic features (spore settling velocity $V_t$ and release height $Z_0$) and extrinsic environmental features (mean horizontal wind speed $\bar{U}$ and canopy height $h$) to generate maps of spatially explicit dispersal kernels. Climatic suitability and dispersal kernel maps, updated at regular intervals, are finally combined to parameterize simulations of dynamic range shifts under changing climatic conditions (Part 3).

snow is an important driver of species distributions in Arctic regions[46], the lack of sufficiently detailed information on snow precipitation across Europe prevented us from implementing this variable.

Given the spatial grain of our study, the hypothesis that some species will persist in small microhabitats, where temperatures can be cooler and humidity higher than in the surrounding environment, cannot be rejected. Data at finer scales for both present and future conditions would therefore be desirable[47]. Recently developed methods to generate fine-grained climatic data taking into account microclimatic effects modulated by microtopographic variation in the terrain, vegetation cover and ground properties using energy balance equations cannot, however, yet be implemented across large spatial scales[48].

For future climate conditions, a wide range of GCMs have been described and their variation represents the largest source of uncertainty in future range prediction studies[49]. No criterion exists to evaluate GCMs, whose performance may vary among regions and variables[50]. Due to computational constrains associated with our migration simulations (see below), we followed Didersky et al.[51]. and selected two GCMs that reflected the highest and lowest levels of predicted changes due to climate change for two angiosperm species in Europe[50], namely MPI-ESM-LR[52] and HadGem2-ES[53]. For each GCM, we analyzed two climate change scenarios. These scenarios are expressed by the representative concentration pathways (RCPs), using values comparing the level of radiative forcing between the preindustrial era and 2100. The moderate scenario RCP4.5 assumes 650 ppm $CO_2$ and 1.0–2.6 °C increase by 2100, and refers to AR4 guideline scenario B1 of IPCC AR4 guidelines. The pessimistic scenario RCP8.5 assumes 1350 ppm $CO_2$ and 2.6–4.8 °C increase by 2100, and refers to A1F1 scenario of IPCC AR4 guidelines[54]. Climatic data for each GCM and each RCP were averaged for each of the four time periods considered, i.e., 2010–2020, 2020–2030, 2030–2040 and 2040–2050.

Monthly average and daily maximum wind speeds measured at 10 m as well as predicted wind speeds for the same ten-year time periods between 2010 and 2050, were computed from EURO-CORDEX (https://euro-cordex.net). Canopy height data were obtained from the global scale mapping of canopy height and biomass at a 1-km spatial resolution[55]. Wind speed and canopy height were sampled for each pixel and each time-slice to generate kernel maps through time (see below).

**Deriving climatic suitability maps.** The correlation among the 19 bioclimatic variables was computed from 50,000 random points. To avoid multicollinearity, five bioclimatic variables with a Pearson correlation value of $R < 0.7$ (as recommended in ref. [30]) were selected. These variables were: mean of monthly temperature range, temperature seasonality, mean temperature of warmest quarter, precipitation of wettest month and precipitation of warmest quarter. Since the geographic background should not only reflect the extant, but also the potentially occupied range in the past[56], and since, in bryophytes, models built from large geographic backgrounds are recommended[57], pseudo-absence points were sampled from a random selection from all points within the studied area excluding available presence points across Europe.

To account for model uncertainty, we generated ensemble models[58] using generalized linear model (GLM)[59] and boosted regression trees (GBM)[60] with the package biomod2 3.3–7[61]. Following Barbet-Massin et al.[62], 10,000 pseudo-absence points were sampled for GLM and then down-weighted to give them same overall prevalence as presences. For GBM, we sampled a number of pseudo-absence points identical to the number of datapoints. For GLM, the default parameter set (selection procedure via AIC, quadratic model, interaction level = 0, interaction level between variables considered, logit function) was used. For GBM, 5000 trees were included, the maximum depth of each tree was set to 5, the fraction of the training set observations randomly selected to propose the next tree in the expansion was set to 0.8. All other parameters were set to default (Bernouilli distribution, minimum number of observations in the terminal nodes of the trees = 5, shrinkage = 0.001, Number of cross-validation folds = 3). Ten replicates were run and, for each run, 80% of the data was used to calibrate the models, whereas the remaining 20% was kept aside to evaluate the performance of the model using the AUC and TSS metrics. We generated a consensus model of the 10 replicates for each of the GLM and GBM models, wherein each individual model contributed proportionally to its goodness-of-fit statistics. Finally, we computed the suitability at each pixel based on the average of the two GLM and GBM consensus models. Because, despite our thorough literature survey to document species distributions, the number of actually sampled points is a dramatic under-estimation of the actual number of occupied pixels by the species across the study area, all pixels identified as climatically suitable by binarized climatic suitability model projections were employed as initial distribution points for migrations during the first time slice.

This could lead to an overestimation of the number of source pixels and raises the issue that, like in any hybrid correlative-mechanistic model, datapoints are employed both for inferring the niche and initiating dispersal simulations, whereas datapoints are themselves the result of a dispersal process[63]. If, due to dispersal constraints, a species is absent from climatically suitable conditions, climatic suitability models may therefore underestimate species niche range[63]. Although bryophytes are extremely good dispersers, so that, unlike in some angiosperm species[18], there is no mismatch between the predicted and observed northern limit of distribution[39], the present analyses suggest that there is a time-lag of more than a century before newly suitable areas are fully colonized. Nevertheless, our datapoints were sampled across the entire European range of the species, encompassing the full range of climatic conditions that they can experience, so that the potential failure to incorporate localities where the species had not time to disperse yet would not affect the boundaries of our global niche estimate.

The ensemble model was then projected onto future climatic layers using two GCMs and two RCPs per GCM (see above). A key issue with modeling responses to climate change is that we do not fully understand how models made under current conditions will transfer to future conditions. Models developed using too many predictors may run the risk of overfitting to local conditions, restricting the predictive power of the model[64,65]. Tests of transferability across taxa and geographic locations have, however, failed to demonstrate consistent patterns, and a general approach to developing transferable models remains elusive[66]. Here, we compared the ROC and maxTSS values computed from the test sets (20% of the data) to those observed at the level of the entire dataset, assuming that these statistics at the level of the entire dataset would substantially drop at the level of the test sets in case of severe issues of overfitting.

The continuous suitability index was transformed into a binary presence/absence model, using maximum TSS to reclassify values.

### Dispersal simulations under changing climate conditions

*The MigClim model.* Dispersal simulations under changing climate conditions were performed with a modified version of Migclim[67] adapted for wind dispersal. To simulate dispersal under climate change, MigClim requires information on species dispersal capacities, a map of species initial distribution, a map of present climate conditions, and maps of future climate conditions at $p$ intervals that divide the period between time present and the end of the simulations, set by the user, into $p$ climatic periods. In MigClim, source pixels are represented by actually occupied pixels and target pixels are pixels that newly become climatically suitable under climate change. Dispersal simulations are performed from source pixels into target pixels as follows (see Fig. 2 in Engler et al.[68]):

1. For each target pixels, all the potential source pixels located within a user-defined range are identified.
2. The probability that the target pixel is colonized from all the potential source pixels is computed through the probability $Pcol$ (see below). Optionally, long-distance can be added to the simulation, with a user-defined range and probability.
3. These steps are repeated $nDisp$ times, with $nDisp$ typically set to 1 year, until the end of the first climatic period.
4. At the end of each of the $p$ climatic periods, pixels that are no longer suitable due to changes in environmental conditions have their values reset to zero, and climatic suitability is updated to reflect environmental change, potentially resulting in a series of newly suitable target pixels.

To define $Pcol$, MigClim implements a dispersal kernel, which is a vector indicating the probability of dispersal $P(x)$ as a function of the distance from the source. Since dispersal from a source pixel could take place in any direction, MigClim implements a coefficient of diffusion called $Surface_j$, which corresponds to the number of pixels belonging to a same distance class from the source, to compute the probability that a diaspore from a source pixel ends-up in a target pixel and not in any other pixel located at the same distance range:

$$P\left(pixel_j\right) = \frac{P(x)}{Surface_j} \qquad (1)$$

To account for the number of diaspores produced by a source pixel $j$, MigClim implements a parameter called Successful Seeds, which accounts for the number of seeds produced by a source pixel $j$ and allows for turning individual dispersal event probabilities into species spread rates.

$$P_{Disp}\left(pixel_j\right) = 1 - \left(1 - P_{Seed}\left(pixel_j\right)\right)^{SuccessfulSeed} \qquad (2)$$

Finally, $P(pixel_j)$ values are computed at increasing distances from the source and combined to generate a global probability of colonization $Pcol$ from $n$ potential source pixels:

$$Pcol = 1 - \prod_{i=1}^{n}\left(1 - P_{disp(i)} \times P_{mat(i)}\right) \qquad (3)$$

where $P_{mat(i)}$ is a probability that is function of the time as the source pixel $I$ became occupied and represents the increase in reproductive potential of source pixel $i$ over time.

### Implementing the Wald model in MigClim for simulating dispersal by wind.

We developed a new version of Migclim, MigClim 1.7[69], designed for wind dispersal. While a single kernel was employed across the landscape until the end of the simulations in the previous implementation of MigClim, we employed a wind-dispersal kernel that was sampled for each pixel individually to account for variations in wind conditions and was modified at the same time as the p climatic change intervals to take future wind conditions into account.

We employed the Wald model[70] to infer dispersal kernels. The WALD model was initially developed[70] and largely used for wind-dispersed seeds[34,35], so that its use for smaller particles could be questioned. Bryophyte spore-trapping experiments in fact revealed that the tail of the dispersal kernel is, beyond hundreds of meters, not distance-dependent, suggesting that, once a spore is airborne, it could disperse over hundreds to thousands of kilometers, regardless of the distance from the source[71]. Spatial genetic structures consistently show, however, significant isolation-by-distance patterns for all distance classes, evidencing that realized colonization rates are distant-dependent[72] and justifying the implementation of a mechanistic model such as WALD. Furthermore, the WALD model assumes that (i) the slippage velocity between the particles and surrounding air is zero, leading to an infinite drag coefficient, so that the particles and surrounding air parcels are tightly coupled, and that (ii) the diaspore terminal velocity is reached instantly after release. These conditions are precisely met in small particles, which (i) are characterized by low Reynolds numbers, and hence, high drag coefficients, and (ii) almost readily reach terminal velocity after release. The WALD model has thus also been applied to small particles such as pollen grains and spores[73,74].

The Wald model[70] defines the probability $P(x)$ of colonization at distance $x$ from the source depending on intrinsic (e.g., settling velocity, height of release) and extrinsic (e.g., wind speed) parameters, across the distance range between the source and target pixels, as follows:

$$P(x) = \sqrt{\frac{\lambda'}{2\pi x^3}}\exp\left(\frac{\lambda'(x - \mu')^2}{2\mu'^2 x}\right) \qquad (4)$$

With $\mu' = \frac{H\bar{U}}{Vt}, \lambda' = \left(\frac{H}{\sigma}\right)^2$ and $\sigma^2 = 2Kh\frac{\sigma_w}{U}$

where $x$ is the distance from the source, $\bar{U}$ is the horizontal mean wind speed at the height of seed release, $H$ is the release height, $h$ accounts for canopy height, Vt is the diaspore terminal velocity, $K$ is von Karman's constant (0.4), and $\sigma_w$ is a turbulence parameter corresponding to the standard deviation of the vertical wind velocity.

Starting from the centroid of a source pixel, we finally integrate the Wald model over the shortest and largest distances between the source and target pixels to obtain the probability of colonization of the latter.

### Parameter estimation.

We derived the turbulence parameter $\sigma_w$ from wind speed data and canopy height[55]. $\sigma_w = 1.25\,u^*$, where $u^*$ is the wind-induced friction velocity depending on canopy height. Since wind speed is typically measured over short vegetation ($h_s$, set at 0.3 m), we first inferred $\sigma_w$ above taller vegetation of variable height h from the wind-induced friction velocity measured above short vegetation, $u_s^*$. Hypothesizing that, at the top of the atmospheric surface layer (~200 m), the mean velocity is not affected by the texture of the ground vegetation,

$$u^* = u_s^*\left(\log(200) - \log\left(\frac{h_s}{10}\right)\right)\Big/\left(\log(200) - \log\left(\frac{h}{10}\right)\right) \qquad (5)$$

Following Nathan et al.[35], $u_s^*$ was estimated using von Karman's formula from the measured wind speed $\bar{U}_s$:

$$\bar{U}_s = \frac{u_s^*\log\left(\frac{w}{Z0_s}\right)}{K} \qquad (6)$$

where $K$ is von Karman's constant (0.4), $w$ is the height, at which the wind was measured (here 10 m), and $Z0_s = 0.1\,h_s$.

The friction velocity $u^*$ for taller vegetation of height h was then derived using Eq. (5).

To derive the mean wind speed at the height of release $H$, we implemented either the logarithmic or exponential wind profile[75]. When the height of release H is roughly higher than the canopy height h, the logarithmic wind profile describes the decline in horizontal wind speed with decreasing height above the surface, due to the surface resistance, as:

$$\bar{U}_H = \frac{u^*\ln\left(\frac{H-d}{Z0}\right)}{K} \qquad (7)$$

with $Z0 = 0.1\,h$ and $d = 0.7\,h$.

In contrast, when the height of release $H$ is below the canopy, we implemented the exponential wind profile:

$$\bar{U}_H = \bar{U}_h\exp\left(\alpha\left(\frac{H}{h} - 1\right)\right) \qquad (8)$$

with the mean wind speed at canopy height $\overline{U_h}$ derived from Eq. (6), and $\alpha$ derived from Gualtieri and Secci[76] as $\alpha = 0.24 + 0.096Z_0 + 0.016\log^2 Z_0$, where $Z_0 = 0.1\,h$

The parameters $\bar{U}$, $h$ and $\sigma_w$ are sampled for each pixel and each time-slice (10 years intervals) to generate kernel maps through time.

We determined "Successful Seed" empirically following the calibration method of Engler and Guisan[67]. Although "Successful Seed" was determined once on the basis of a single empirical study[71] and kept constant across species, this study reported observed colonization rates at distances of hundreds of meters from the source colony, giving us a unique opportunity to make the link between our deterministic models and actual observations, increasing the realism of our approach. $P_{mat}$ was set to 1.

Finally, in addition to short-distance dispersal events with a probability defined by the kernel described above, any pixel located at >10 km from a potential source could be colonized by LDD. The maximum LDD distance was set to unlimited based on phylogeographic evidence[39]. Following Robledo-Arnuncio et al.[31], we employed the results of previous Approximate Bayesian Computation methods for LDD inference from genetic structure data in bryophytes[39,77] to define the range of LDD probability values, set to 0, $10^{-4}$, $10^{-3}$, $10^{-2}$ and $10^{-1}$.

**Migclim simulations**. We modeled the dispersal of a species under a climate change scenario over a period of 40 years, from 2010 to 2050. Starting with an initial distribution for the year 2010, the climatic suitability of cells was updated every 10 years to reflect the projected changes in climatic conditions under the considered climate change scenario. Since our simulations run over 40 years, we need four different climatic suitability maps. The wind layers were updated at the same 10 years intervals as the climatic data to produce series of spatially and temporally explicit kernel maps. We assume that our species disperse once a year, and hence, our simulations performed a total of 40 dispersal steps between 2010 and 2050. For each 10 years climatic period, pixels were identified as potentially suitable based on the binarized climatic suitability model projections. While climatic suitability thus drove colonization probability, a recent study raised the intriguing idea that spread rates at the migration front increase as climatic suitability decreases as a response to the need to seek for more suitable habitats[78]. In bryophytes, however, such a mechanism would be unlikely as inadequate resources and investment in environmental stress defence typically result in shifts from sexual to asexual reproduction[79].

For each species, we ran a sensitivity analysis by testing the impact of variation of the free parameters described above: two values of horizontal windspeed $\bar{U}$ (monthly average and daily maximum), three values of spore release height $Z_0$ (0.03, 1 and 10 m), and four values of LDD probabilities (see above). For each parameter combination, 30 MigClim replicates were performed.

We computed the ratio between the predicted loss of suitable area (fraction of initially suitable cells that became unsuitable by 2050) and the simulated effective colonization rate (fraction of newly suitable cells by 2050 that were effectively colonized) using two extreme values of the LDD probability range, that is, 0 and 0.1.

To determine the time-lag of the colonization of newly suitable habitats, the analyses were run for 500 years, keeping the environmental parameters at their 2050 values.

**Reporting summary**. Further information on research design is available in the Nature Research Reporting Summary linked to this article.

## Data availability

Occurrence data are available from Figshare (https://doi.org/10.6084/m9.figshare.8289650).

## Code availability

Migclim 1.7 and all the R scripts for the analyses presented here are available on GitHub[69].

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

## Acknowledgements

Many thanks are due to the National Data and Information Center on Swiss Bryophytes ("Swissbryophytes") for sharing data and to G. Katul for his assistance with the implementation of the WALD model. F.C. was funded by the FRIA grant of F.R.S.-FNRS (grant no. 5105819F). Computational resources were provided by the Fédération Wallonie-Bruxelles (Tier-1; funded by Walloon Region, grant no. 1117545), the Consortium des Équipements de Calcul Intensif (CÉCI; funded by the F.R.S.-FNRS, grant no. 2.5020.11), and through two grants to DB (University of Liège". Crédit de démarrage 2012" SFRD-12/04; F.R.S.- FNRS "Crédit de recherche 2014" CDR J.0080.15).

## Author contributions

A.G., A.V., F.Z. and R.E. designed the framework of the study. A.V., B.P., J.M. and R.G.M. collected the data. D.B., F.C., F.Z., O.B. and R.E. performed the analyses. A.V., F.C. and F.Z. wrote the paper with the assistance from all co-authors.

## Competing interests

The authors declare no competing interests.
