## [Peer Review File · Nature Communications]

Reviewers' Comments:

Reviewer #1:

Remarks to the Author:

This is an interesting study which uses explicit modelling of wind dispersal, combined with projected future wind speeds and climate suitability to 2050 to project distributions of 10 bryophytes in Europe. I am generally happy with the modelling used, although I have a number of comments. I would generally see it as a useful addition to the literature, rather than the game-changer implied by the authors. I explain myself below.

The explicit modelling of dispersal in projecting future distributions of species is important. In fact, this has long been acknowledged as critical to species distribution modelling(1) and specifically for responses to climate change(2). So, it is rather odd for the authors to claim this as an area of controversy (line 21). I am not being merely pedantic here, because the paper implies a level of novelty in the approach, which is simply not the case. It is stated that "no study has, to date, attempted at predicting range shifts in an explicit dispersal framework at large, continental scale" (lines 66-68). Several papers have done so(3-6), and others have done so over smaller areas, e.g.(7,8). It is interesting that the authors do mention a couple of these papers(5,7), but right at the end of the paper, almost as an afterthought. While novelty is not vital, I do not see how the approach here moves the field on greatly from these other studies, especially given some of the drawbacks I mention below. Connected to this issue is the title. The paper does not really explain why bryophytes are canaries in the climate warming coal mine, which implies they provide an early warning of either climate change or its impacts. So, I think the title is misleading.

Any modelling involves simplifications, but some are debatable and at least need better justification. In my comments below, I asterisk the major issues. But I would say that the study ultimately gives some well-known results: spread has a high sensitivity to windspeed and dispersule (spore) size and there is strong dispersal limitation so that colonisation is delayed.

1) *Windspeed is derived at 10m height. As the vertical wind profile is logarithmic, the amounts to a vast over-estimate of wind speed at spore release height. The authors acknowledge this (line 289-290), but do not implement well-known methods to correct the estimate; for example ref(9).

2) As the species will generally need to spread northwards, wind direction is important. As elsewhere, wind directions across Europe are not isotropic as assumed in this modelling. GCMs provide wind direction projections. I understand this may be too much to ask for this modelling, but it is worth some discussion at least.

3) It is unclear whether the time slices of the climate projections are 10 year or 1 year intervals (line 295).

4) *Only one GCM is used (line 342). The authors imply the one used is "recommended for Europe" (line 343-344). I could not find this recommendation in the cited papers. As GCMs vary so much, especially in windspeed projections (e.g.5,7), this is a major assumption. The CMIP5 data represents a large number of models. Ideally, the authors should consider >1 GCM. At the least, they need to caveat this study as being representative of a specific GCM, which means their projections cannot be seen as very representative.

5) Similarly, a better justification of why RCPs 4.5 and 8.5 are used. This seems reasonable, but needs better explanation.

6) *MigCLIM is never well described. It is a long-standing model. I have no objection to it being used, but this does undermine the authors claim for novelty. It does have limitations however, in that colonisation is a simple transition function without representing demography. In particular how demography might vary with climate. As the authors admit, they use one study to estimate this function (line 387). This parameter is constant across all species and is not subject to sensitivity analysis. Again, this undermines how representative we can view this study.

7) Similarly, the WALD model is poorly described. For example, how is vegetation roughness (& so wind turbulence) parameterised?

- dichotomy. *J. Biogeogr.* 39, 2119-2131, doi:10.1111/j.1365-2699.2011.02659.x (2012).
- 2 Corlett, R. T. & Westcott, D. A. Will plant movements keep up with climate change? *Trends Ecol. Evol.* 28, 482-488, doi:https://doi.org/10.1016/j.tree.2013.04.003 (2013).
- 3 Sirois-Delisle, C. & Kerr, J. T. Climate change-driven range losses among bumblebee species are poised to accelerate. *Sci. Rep.* 8, 10, doi:10.1038/s41598-018-32665-y (2018).
- 4 Dullinger, S. et al. Modelling the effect of habitat fragmentation on climate-driven migration of European forest understorey plants. *Divers. Distrib.* 21, 1375-1387, doi:10.1111/ddi.12370 (2015).
- 5 Nathan, R. et al. Spread of North American wind-dispersed trees in future environments. *Ecol. Lett.* 14, 211-219, doi:10.1111/j.1461-0248.2010.01573.x (2011).
- 6 Santini, L. et al. A trait-based approach for predicting species responses to environmental change from sparse data: how well might terrestrial mammals track climate change? *Glob. Chang. Biol.* 22, 2415-2424, doi:10.1111/gcb.13271 (2016).
- 7 Bullock, J. M. et al. Modelling spread of British wind-dispersed plants under future wind speeds in a changing climate. *J. Ecol.* 100, 104-115, doi:10.1111/j.1365-2745.2011.01910.x (2012).
- 8 Engler, R. & Guisan, A. MigClim: Predicting plant distribution and dispersal in a changing climate. *Divers. Distrib.* 15, 590-601, doi:10.1111/j.1472-4642.2009.00566.x (2009).
- 9 Skarpaas, O. & Shea, K. Dispersal patterns, dispersal mechanisms, and invasion wave speeds for invasive thistles. *American Naturalist* 170, 421-430, doi:10.1086/519854 (2007).

Reviewer #2:

Remarks to the Author:

Bryophytes are the canaries in the coal mine of climate warming

Zanatta F, Engler R, Collart F, Broenimann O, Mateo R.G, Papp B, Muñoz J, Baurain D, Guisan A. & Vanderpoorten A.

The authors seek to investigate the extent to which species have the ability to balance the loss of suitable habitats due to climate warming by shifting their ranges, by considering a group that are extremely efficient dispersers - the bryophytes. This is a vitally important topic and the bryophytes are a great group to examine this in. However, in considering if these plants can indeed track shifts in areas of suitable climate it is important to ensure that the habitat one is tracking is relevant to group under consideration. In the case of bryophytes this is especially important since they are very small plants and thus their habitat likely consists of very small niches within a broader landscape.

In recognition of this, several recent papers have examined microhabitat at somewhat smaller scales and to consider what this means for future climate models and future diversity. Their findings suggest it is important for most biodiversity, both in terms of explaining where things are now and where they will be in future. As well as for understanding ecosystem functioning I am including quotes from two examples.

"Many questions in global change ecology deal with large-scale patterns, with global databases of species distributions, species traits and ecosystem processes becoming increasingly available (e.g. Bruelheide et al., 2018). Current analyses of these large-scale patterns— and their predictions under anthropogenic climate change— often rely on spatially coarse-resolution global climatic data interpolated from weather station measurements. These weather stations are systematically located in open landscapes, where the wind continuously mixes the air, and are shielded from direct solar radiation, thus ignoring many climate-forcing processes that operate near the ground, at very fine spatial resolutions, and in microhabitats that vary in their terrain, exposure to winds and vegetation cover (De Frenne & Verheyen, 2016). Importantly, while these microclimatic processes often operate at fine spatial resolutions, they do affect broad-scale ecological processes like species distributions and ecosystem functioning." From Lambrechts (2019; DOI: 10.1111/gcb.14942)

"Most studies on the biological effects of future climatic changes rely on seasonally aggregated, coarse-resolution data. Such data mask spatial and temporal variability in microclimate driven by terrain, wind and vegetation, and ultimately bear little resemblance to the conditions that organisms experience in the wild."

"High-temporal resolution extremes in conditions under future climate change were predicted to be considerably less novel than the extremes estimated using seasonally aggregated variables. The study highlights the need to more accurately estimate the future climatic conditions experienced by organisms." Maclean (2019 DOI: 10.1111/gcb.14876)

The technical advance from Maclean 2019 actually provides a model that can be used to predict future climate at somewhat higher spatial and temporal resolution. Both these papers also refer to other literature which may be of use to the authors.

My question for the authors of this manuscript is, can they demonstrate that the 1 km² pixels that they are working with adequately represent the microclimate for Sphagnum? I do appreciate that Sphagnum is a giant in the bryophyte world but even so, is the appropriate scale for considering its future climatic niches really 1 km². I think this paper would be far more relevant if the authors can address this issue of microclimate variability and the relevance that this has for the future modelling of Sphagnum biodiversity. I appreciate that redoing the modelling is probably not possible but I think it is very important to recognize and reflect on the mismatch in spatial scales.

Additional comments

P3 line 77 I do not think the statement "Second, bryophytes exclusively rely on rainfall for water uptake and exhibit lower tolerance to warm temperatures than angiosperms²³," is accurate. It depends on the angiosperms and the bryophytes species you are considering. Many bryophytes survive in hot deserts. Also in relation to rainfall, I think you mean that the bryophytes are not obtaining water from the soil via roots. They might get water from melt streams and lakes, bogs, snow etc. As it is written it implies that all their water comes directly as rainfall.

P13 line 277 "We restricted our sampling to data collected after 1960 because the latter often lack sufficiently precise location information and to avoid a potential mismatch between old observations and current climate conditions used for modelling."

I can appreciate the reasoning behind this , but how much data was collected pre 1960 versus since? Does this have the potential to have skewed the data? I imagine that there might have been a lot more historic sampling of bryophytes.

P14 line 307 How well does the climatic data account for water availability in regions where a large proportion of plant water comes from snow fall (precipitation confounded by blowing snow, sublimation etc) and melt water from glaciers?

Reviewer #3:

Remarks to the Author:

Review of NCOMMS-19-39281 "Bryophytes are the canaries in the coal mine of climate warming"

In this work a process-oriented model of areas of distribution of bryophyte species is presented, in the form of a cellular automaton describing the dynamics of the distributional ranges of the species in question, as governed by niche requirements, dispersal of spores, and climate change. The work is interesting and relevant, and it is indeed one of the first examples of which I am aware in which climate change is allowed to force the parameters of a set of process-oriented distribution models.

I have comments on several points, however, none of which is serious.

1. The methodology of what is in fact a relatively complicated simulation is very schematic, inconsistent or doubtful. For instance:

a. the parametrization of MigClim is inconsistent with the published literature on this package. In Line 353, is $P(\text{pixel } j)$ the same as P_{col} in eq. (1) of Engler (2012)? If so, P_{inv} in that equation (1) will be calculated from the output of BioMod? These points need to be made explicit and clear. In L 398, is $nDisp$ the same as $dispSteps$ in Engler (2012)? In general it would be very good for the reader if authors would maintain the same symbols as in previous literature, or to explicitly state which ones have been changed. Maybe adding a table with the parameters and how they were obtained would be a good idea.

b. Another methodological point that needs clarification is the use of so called "pseudoabsences" (L311). Which one of the at least four types (Barbet-Massin et al. 2012) of possible "pseudoabsences" is used? They are by no means equivalent, and using one or the other may change radically the interpretation of the output of a regression method, from a true probability to a relative probability, and from being able to use rigorously an AUC to better avoiding it. Be explicit about which type is being used.

c. The authors chose to model environmental tolerances using five bioclimatic variables from WorldClim. Then they used BioMod to obtain what, strictly speaking, are response curves to climatic variables (also called ecological niches, and I assume these are used to parametrize P_{inv} in equation (1), if this is the case, please say it explicitly). Now, they use GLIM and GBM as their model algorithms. These, mostly the GBM, and with five predictors, can result in "data hugging", meaning that they will be faithful models of the observations, but not necessarily good for transferring predictions. However, this paper is very much about transferring. Would not be preferable to fit fewer variables and simpler shapes to the data points? Please discuss this matter in a couple of lines.

d. It is not at all clear how LDD is calculated: "Here, LDD was implemented beyond the range covered by our dispersal simulations, i.e., beyond 10km" OK, but how much is "beyond"? 0.01 km? 1,000 km? Then, in L 410 you state that you use "... two extreme values of the probability range of LDD, that is, 0 and 0.1" Are 0 and 0.1 "extreme values"? This needs explanation. It would add much clarity having a detailed description of how LDD are actually estimated.

2. Finally, my personal obsession. In line 297 you call "Species distribution models" the section where you are calculating the suitability of cells (L323-324): "Finally, we computed the suitability at each pixel based on the average of the two GLM and GBM consensus models." In this section you are modeling suitability, NOT distributions. You will be modeling the actual distribution when you add dispersal, (clearly assuming interactions do not matter). What you do in the SDM section is actually to estimate one parameter (P_{inv}), the local suitability (also called the niche), for the simulation that actually estimates the distribution. Why to call the section "Species distribution models" when you are modeling climatic suitability (not "habitat suitability", as stated in fig 4. and L396, you are calculating climatic suitability)? The distinction matters. In L62-65 you state that adding dispersal adds predictive power to models. Of course it does! Simple correlative models are not SDMs, but ENMs projected to geography, and this projection models area of suitability (potential distributions). To get the actual distribution more factors are needed, like interactions and movements. However, if you fail to distinguish between a potential distribution and an actual one confusion follows. Maybe have another look at Fordham et al. (2017) to see how they distinguish between modeling distributions, and modeling niches. Now, since a large chunk of the literature does not bother with distinguishing between modeling something in geography (a distribution, with several types) and something in an abstract multidimensional space (the limits of tolerance, or "niches"), I am not going to ask that you change your terminology. I simply suggest you consider the possibility.

3. The paper does not cite several quite relevant references. Maybe the journal imposes a limit to the number of references, but I suggest that the authors consider citing the papers I will suggest below.

4. Minor points.

- a. L 81. What are "intraspecific variations in wind conditions"?
- b. L. 189-190, you state that your approach "... over-estimate colonization rates because all climatically suitable pixels for the present time served as starting points in dispersal simulations," Well, in the first place, you are surely not talking about overestimating "rates" but overestimating the result of the simulation. In the second place, if you are modeling the future distribution of a species, you start your simulation from where it exists today. I think what you did is right, starting the spread from every recorded locality, since you are asking questions about future distributions considering movements. The occupied localities are the initial conditions for the cellular automaton.
- c. L. 212, "...under the most optimistic dispersal conditions" What are "most optimistic"? One needs to see the parameters...
- d. L 204-205. You state that wider distribution ranges imply wider climatic niches. Not necessarily. Imagine a species widely distributed over northern Russia, with a very narrow climatic niche. Or the opposite, a species with a wide range of climatic tolerances, endemic to a single island. This statement is not convincing.
- e. L. 392, by "integrated" you mean summed over pixels?

Bibliography suggested

Colwell, R. K., and T. F. Rangel. 2010. A stochastic, evolutionary model for range shifts and richness on tropical elevational gradients under Quaternary glacial cycles. *Philosophical Transactions of the Royal Society B: Biological Sciences* 365:3695-3707.

Hooten, M. B., and C. Wikle. 2008. A hierarchical Bayesian non-linear spatio-temporal model for the spread of invasive species with application to the Eurasian Collared-Dove. *Environmental and Ecological Statistics* 15:59-70.

Ingenloff, K., C. M. Hensz, T. Anamza, V. Barve, L. P. Campbell, J. C. Cooper, E. Komp et al. 2017. Predictable invasion dynamics in North American populations of the Eurasian collared dove *Streptopelia decaocto*. *Proceedings of the Royal Society B: Biological Sciences* 284:20171157.

Qiao, H., E. E. Saupe, J. Soberón, A. T. Peterson, C. E. Myers, D. C. Collar, and J. L. Bronstein. 2016. Impacts of Niche Breadth and Dispersal Ability on Macroevolutionary Patterns. *The American Naturalist* 188:149-162.

Schurr, F. M., J. Pagel, J. S. Cabral, J. Groeneveld, O. Bykova, R. B. O'Hara, F. Hartig et al. 2012. How to understand species' niches and range dynamics: a demographic research agenda for biogeography. *Journal of Biogeography* 39:2146-2162.

Reviewer #1 (Remarks to the Author):

This is an interesting study which uses explicit modelling of wind dispersal, combined with projected future wind speeds and climate suitability to 2050 to project distributions of 10 bryophytes in Europe. I am generally happy with the modelling used, although I have a number of comments. I would generally see it as a useful addition to the literature, rather than the game-changer implied by the authors. I explain myself below.

>> Thanks for the overall positive comments on the modelling and analyses, and for the useful critical comments that helped improve the ms further. We acknowledge the need to better explain the novelty brought by the study, which we did as we explain below in response to your specific comments.

The explicit modelling of dispersal in projecting future distributions of species is important. In fact, this has long been acknowledged as critical to species distribution modelling(1) and specifically for responses to climate change(2). So, it is rather odd for the authors to claim this as an area of controversy (line 21).

>> We fully agree that the modelling of dispersal per se is by far not new, and its importance already attested, and we acknowledge that the controversy we mention was perhaps not sufficiently explained. We mean here that while the need to include dispersal into predictive models of species distributions has indeed been long acknowledged, there is still a debate about the ability of species in different groups to track areas of suitable climate. In this manuscript, we address this question in expectedly extremely efficient dispersers. We have now better explained this in the introduction, and thank you for the additional references provided, which allowed us to better document this point.

I am not being merely pedantic here, because the paper implies a level of novelty in the approach, which is simply not the case. It is stated that “no study has, to date, attempted at predicting range shifts in an explicit dispersal framework at large, continental scale” (lines 66-68). Several papers have done so(3-6), and others have done so over smaller areas, e.g.(7,8). It is interesting that the authors do mention a couple of these papers(5,7), but right at the end of the paper, almost as an afterthought. While novelty is not vital, I do not see how the approach here moves the field on greatly from these other studies, especially given some of the drawbacks I mention below.

>> Thank you for your comment and the references, which allowed us to better set the present study in the right context and explain how it compares with previous studies. We agree that some references were coming too late in the ms, and we now discuss them earlier, in the introduction. Programs for simulating range shifts in a spatially-explicit framework are available (e.g. Cats, Dullinger et al. 2012, Shift, Prasad et al. 2013, RangeShifter, Bocedi et al. 2014). Using these methods, studies on range shifts as a response to warming at a large scale have been published (Meier et al. 2012, Dullinger et al. 2012, Prasad et al. 2013, Dullinger et al. 2015). However, in these studies, a dispersal kernel that is constant through the landscape was employed, or actual migration rates were randomly sampled from a prior distribution. Here we focused on wind dispersal, which is strongly impacted by local variations of wind speed and canopy height, requiring a spatially-explicit approach of the dispersal kernel. Much more complex models than ours (integrating for instance demographic parameters) were implemented, but across very short ranges (Nathan et al. 2011, Bullock et al. 2012). It therefore appears to us that the originality of the present study is hence that it is, to our knowledge, the first one that attempts at modelling variation in environmental conditions and dispersal at a large scale in a spatially explicit context, where dispersal itself is constrained by pixel-specific features of wind speed and canopy height. We have now modified the introduction and discussion accordingly.

Connected to this issue is the title. The paper does not really explain why bryophytes are canaries in the climate warming coal mine, which implies they provide an early warning of either climate change or its impacts. So, I think the title is misleading.

>> We agree and modified the title for: ‘Even bryophytes lag behind climate change’.

Any modelling involves simplifications, but some are debatable and at least need better justification. In my comments below, I asterisk the major issues. But I would say that the study ultimately gives some well-known results: spread has a high sensitivity to windspeed and dispersule (spore) size and there is strong dispersal limitation so that colonisation is delayed.

1) *Windspeed is derived at 10m height. As the vertical wind profile is logarithmic, the amounts to a vast over-estimate of wind speed at spore release height. The authors acknowledge this (line 289-290), but do not implement well-known methods to correct the estimate; for example ref(9).

>> We agree and thank you for this comment, which helped us improving our approach. We now calculate wind speed at release height differently when the release is below or above the canopy. The description of the approach now reads; “To derive the mean wind speed at the height of release H, we implemented either the logarithmic or exponential wind profile method (Nathan et al. 2002). When the height of release H is roughly higher than the canopy height h, the logarithmic wind profile describes the decline in horizontal wind

speed with decreasing height above the surface, due to the surface resistance, as: $\bar{U}_H = \frac{u^* \ln(\frac{H-d}{Z_0})}{\kappa}$ (6), with $Z_0=0.1h$ and $d=0.7h$

In contrast, when the height of release H is below the canopy, we implemented the exponential wind profile:

$$\bar{U}_H = \bar{U}_h \exp\left(\alpha \left(\frac{H}{h} - 1\right)\right)$$

with the mean wind speed at canopy height \bar{U}_h derived from eq. 6, and α derived from Gualtieri & Secci (2011) as $\alpha = 0.24 +$

$0.096 Z_0 + 0.016 \log^2 Z_0$, where $Z_0=0.1h$

We described the approach in the M&M section and re-ran all the simulations based on this.

2) As the species will generally need to spread northwards, wind direction is important. As elsewhere, wind directions across Europe are not isotropic as assumed in this modelling. GCMs provide wind direction projections. I understand this may be too much to ask for this modelling, but it is worth some discussion at least.

>> **We agree and mentioned the limitations of our approach in the discussion.**

3) It is unclear whether the time slices of the climate projections are 10 year or 1 year intervals (line 295).

>> >> **Sorry if this was not clear. Both the climate and wind layers were changed at the same 10 yr intervals. We have now made it explicit in the revised M&M section.**

4) *Only one GCM is used (line 342). The authors imply the one used is “recommended for Europe” (line 343-344). I could not find this recommendation in the cited papers. As GCMs vary so much, especially in windspeed projections (e.g.5,7), this is a major assumption. The CMIP5 data represents a large number of models. Ideally, the authors should consider >1 GCM. At the least, they need to caveat this study as being representative of a specific GCM, which means their projections cannot be seen as very representative.

>> **We agree and ran a new set of analyses including a second GCM, which is now fully described in the revised version. We chose HadGem2 because it is the one that differs most from MPI-ESM-LR that we initially implemented. Implementing additional GCMs would of course be desirable to take uncertainty of future climate change into account but was impossible here due to computational constraints. We accordingly modified the paragraph related to this issue to justify our approach as follows:**

‘A wide range of Global Circulation Models (GCMs) have been described and their variation represent the largest source of uncertainty in future range prediction studies (Steen et al. 2017). No criterion exists to evaluate GCMs, whose performance may vary among regions and variables (Goberville et al. 2015). Due to computational constraints associated with our migration simulations (see below), we followed Didersky et al. (2017) and selected two GCMs that reflected the highest and lowest levels of predicted changes due to climate change for two angiosperm species in Europe (Goberville et al. 2015), namely MPI-ESM-LR (Giorgetta et al. 2013) and HadGem2-ES (Jones et al. 2011).’

5) Similarly, a better justification of why RCPs 4.5 and 8.5 are used. This seems reasonable, but needs better explanation.

>> **We agree and expanded the relevant paragraph in the M&M section as follows:**

‘For each GCM, we analyzed two climate change scenarios. These scenarios are expressed by the representative concentration pathways (RCPs), using values comparing the level of radiative forcing between the preindustrial era and 2100. The moderate scenario RCP4.5 assumes 650 ppm CO₂ and 1.0–2.6°C increase by 2100, and refers to AR4 guideline scenario B1 of IPCC AR4 guidelines. The pessimistic scenario RCP8.5 assumes 1,350 ppm CO₂ and 2.6–4.8°C increase by 2100, and refers to A1F1 scenario of IPCC AR4 guidelines (Harris et al., 2014).’

6) *MigCLIM is never well described. It is a long-standing model. I have no objection to it being used, but this does undermine the authors claim for novelty. It does have limitations however, in that colonisation is a simple transition function without representing demography. In particular how demography might vary with climate. As the authors admit, they use one study to estimate this function (line 387). This parameter is constant across all species and is not subject to sensitivity analysis. Again, this undermines how representative we can view this study.

>> **We agree and thank you for your comment, which helped us improving our manuscript to make it clear what was actually modified in MigClim to perform the present study. We therefore divided the relevant sections of the M&Ms to (i) provide a detailed description of MigClim, (ii) explain why and how MigClim was modified to make it work for wind-dispersed species – which required the implementation of a wind-dispersal kernel (the Wald model), whose wind parameters (wind velocity and canopy height) were sampled for each pixel individually instead of using a constant dispersal kernel across the landscape as implemented in the previous version of the program. Our approach does make important assumptions, which we discuss in the revised version. Most importantly, however, these assumptions result in an over-rather than an under-estimation of colonization rates, so that our approach is conservative in the sense that, as in Dullinger et al. (2015), they should be at the upper bound of those achievable. This is also better stressed now in the ms.**

7) Similarly, the WALD model is poorly described. For example, how is vegetation roughness (& so wind turbulence) parameterised?

>> **In line with your previous comment, we added two paragraphs in the M&M section to describe the Wald model and explain how its parameters were derived in the present study. Specifically, we derived the turbulence parameter σ_w from wind speed data³³ and canopy height³⁹. $\sigma_w = 1.25 u^*$, where u^* is the wind-induced friction velocity depending on canopy height. Since wind speed in Ref33 are typically measured over short vegetation (h_s , set at 0.3m), we first inferred σ_w above taller vegetation of variable height h from the wind-induced friction velocity measured above short vegetation, u_s^* . Hypothesizing that, at the top of the atmospheric surface layer (~200m), the mean velocity is not affected by the texture of the ground vegetation,**

$$u^* = u_s^* (\log(200) - \log(\frac{h_s}{10})) / (\log(200) - \log(\frac{h}{10})) \quad (5)$$

Following Nathan et al. (2011), u_s^* was estimated using von Karman’s formula from the measured wind speed \bar{U}_s :

$$\bar{U}_s = \frac{u_s^* \log(\frac{w}{Z0_s})}{K} \quad (6)$$

where K is von Karman’s constant (0.4), w is the height, at which the wind was measured (here 10m), and $Z0_s = 0.1 h_s$. The friction velocity u^* for taller vegetation of height h was then derived using eq. 5.

1 Dormann, C. F. et al. Correlation and process in species distribution models: bridging a dichotomy. *J. Biogeogr.* **39**, 2119-2131, doi:10.1111/j.1365-2699.2011.02659.x (2012).

2 Corlett, R. T. & Westcott, D. A. Will plant movements keep up with climate change? *Trends Ecol. Evol.* **28**, 482-488, doi:https://doi.org/10.1016/j.tree.2013.04.003 (2013).

3 Sirois-Delisle, C. & Kerr, J. T. Climate change-driven range losses among bumblebee species are poised to accelerate. *Sci. Rep.* **8**, 10, doi:10.1038/s41598-018-32665-y (2018).

4 Dullinger, S. et al. Modelling the effect of habitat fragmentation on climate-driven migration of European forest understorey plants. *Divers.*

Distrib. 21, 1375-1387, doi:10.1111/ddi.12370 (2015).

5 Nathan, R. et al. Spread of North American wind-dispersed trees in future environments. *Ecol. Lett.* 14, 211-219, doi:10.1111/j.1461-0248.2010.01573.x (2011).

6 Santini, L. et al. A trait-based approach for predicting species responses to environmental change from sparse data: how well might terrestrial mammals track climate change? *Glob. Chang. Biol.* 22, 2415-2424, doi:10.1111/gcb.13271 (2016).

7 Bullock, J. M. et al. Modelling spread of British wind-dispersed plants under future wind speeds in a changing climate. *J. Ecol.* 100, 104-115, doi:10.1111/j.1365-2745.2011.01910.x (2012).

8 Engler, R. & Guisan, A. MigClim: Predicting plant distribution and dispersal in a changing climate. *Divers. Distrib.* 15, 590-601, doi:10.1111/j.1472-4642.2009.00566.x (2009).

9 Skarpaas, O. & Shea, K. Dispersal patterns, dispersal mechanisms, and invasion wave speeds for invasive thistles. *American Naturalist* 170, 421-430, doi:10.1086/519854 (2007).

Reviewer #2 (Remarks to the Author):

Bryophytes are the canaries in the coal mine of climate warming

Zanatta F, Engler R, Collart F, Broenimann O, Mateo R.G, Papp B, Muñoz J, Baurain D, Guisan A. & Vanderpoorten A.

The authors seek to investigate the extent to which species have the ability to balance the loss of suitable habitats due to climate warming by shifting their ranges, by considering a group that are extremely efficient dispersers - the bryophytes. This is a vitally important topic and the bryophytes are a great group to examine this in. However, in considering if these plants can indeed track shifts in areas of suitable climate it is important to ensure that the habitat one is tracking is relevant to group under consideration. In the case of bryophytes this is especially important since they are very small plants and thus their habitat likely consists of very small niches within a broader landscape.

In recognition of this, several recent papers have examined microhabitat at somewhat smaller scales and to consider what this means for future climate models and future diversity. Their findings suggest it is important for most biodiversity, both in terms of explaining where things are now and where they will be in future. As well as for understanding ecosystem functioning I am including quotes from two examples.

“Many questions in global change ecology deal with large-scale patterns, with global databases of species distributions, species traits and ecosystem processes becoming increasingly available (e.g. Bruelheide et al., 2018). Current analyses of these large-scale patterns—and their predictions under anthropogenic climate change—often rely on spatially coarse-resolution global climatic data interpolated from weather station measurements. These weather stations are systematically located in open landscapes, where the wind continuously mixes the air, and are shielded from direct solar radiation, thus ignoring many climate-forcing processes that operate near the ground, at very fine spatial resolutions, and in microhabitats that vary in their terrain, exposure to winds and vegetation cover (De Frenne & Verheyen, 2016). Importantly, while these microclimatic processes often operate at fine spatial resolutions, they do affect broad-scale ecological processes like species distributions and ecosystem functioning.” From Lambrechts (2019; DOI: 10.1111/gcb.14942)

“Most studies on the biological effects of future climatic changes rely on seasonally aggregated, coarse-resolution data. Such data mask spatial and temporal variability in microclimate driven by terrain, wind and vegetation, and ultimately bear little resemblance to the conditions that organisms experience in the wild.”

“High-temporal resolution extremes in conditions under future climate change were predicted to be considerably less novel than the extremes estimated using seasonally aggregated variables. The study highlights the need to more accurately estimate the future climatic conditions experienced by organisms.” Maclean (2019 DOI: 10.1111/gcb.14876)

The technical advance from Maclean 2019 actually provides a model that can be used to predict future climate at somewhat higher spatial and temporal resolution. Both these papers also refer to other literature which may be of use to the authors.

My question for the authors of this manuscript is, can they demonstrate that the 1 km² pixels that they are working with adequately represent the microclimate for Sphagnum? I do appreciate that Sphagnum is a giant in the bryophyte world but even so, is the appropriate scale for considering its future climatic niches really 1 km². I think this paper would be far more relevant if the authors can address this issue of microclimate variability and the relevance that this has for the future modelling of Sphagnum biodiversity. I appreciate that redoing the modelling is probably not possible but I think it is very important to recognize and reflect on the mismatch in spatial scales.

>> Thank you for this insightful comment and for the suggestions of references and associated findings. We acknowledge the importance to reflect on any potential scale mismatch and neglect of micro-habitat effects. It has indeed long been acknowledged (e.g. Anderson 1963. *The Bryologist* 66, 107–119) that small organisms like bryophytes may occupy micro-habitats where they may persist after that macroclimatic conditions have changed. In these conditions, the method described by Mclean is a great avenue for research. It cannot, however, be currently implemented in large-scale studies like the present one due to computational restrictions. As Mclean indicates (4th paragraph of the discussion), ‘It should be acknowledged, however, that the approach presented here is only feasible over relatively small regions. Accurate representation of global or regional climate at high spatio-temporal resolution is impractical, even with rapid advances in computer processing power and high-resolution remote sensing data’. We further suggest that 1km² is a fairly accurate grain given the large spatial scale of the present study, sufficient to build robust species distribution models at this scale, as evidenced by the goodness-of-fit statistics of these models. We agree this was perhaps not sufficiently justified, and we have now added the following paragraph in the revised ms: ‘Given the spatial grain of our study, the hypothesis that some species will persist in small microhabitats, where temperatures can be cooler and humidity higher than in the surrounding environment, cannot be rejected. Data at finer scales for both present and future conditions would therefore be desirable (Lembrechts & Lenoir 2020). Recently developed methods to generate fine-grained climatic data taking into account microclimatic effects modulated by microtopographic variation in the terrain, vegetation cover and ground properties using energy balance equations cannot, however, yet be implemented across large spatial scales (McLean 2020).’

Additional comments

P3 line 77 I do not think the statement “Second, bryophytes exclusively rely on rainfall for water uptake and exhibit lower tolerance to warm temperatures than angiosperms²³,” is accurate. It depends on the angiosperms and the bryophytes species you are considering. Many bryophytes survive in hot deserts. Also in relation to rainfall, I think you mean that the bryophytes are not obtaining water from the soil via roots. They might get water from melt streams and lakes, bogs, snow etc. As it is written it implies that all their water comes directly as rainfall.

>> Thank you for your comment, which allows us to be more specific. There are indeed (few) bryophyte species that can tolerate extreme heat, but the point that we wanted to make here is that ‘bryophytes show a tendency to be highly sensitive to elevated temperatures’ (He et al. 2016). Furness & Grime (1981) demonstrated, for instance, that if one considers a temperate flora of angiosperm and bryophyte species, bryophytes species exhibit lower temperature optima than their angiosperm counterpart. We also modified the sentence related to water uptake following your recommendations.

P13 line 277 “We restricted our sampling to data collected after 1960 because the latter often lack sufficiently precise location information and to avoid a potential mismatch between old observations and current climate conditions used for modelling.”

I can appreciate the reasoning behind this, but how much data was collected pre 1960 versus since? Does this have the potential to have skewed the data? I imagine that there might have been a lot more historic sampling of bryophytes.

>> This is a good point and we computed the proportion of pre-1960 data in GBIF for our target species to address it. Pre-1960 data represented, on average, 41±12% of the pre- and post-1960 data available (excluding data without any date). Thus, we lost about 40% of the data by excluding pre-1960 records. We rephrased the relevant section of the M&Ms as follows: ‘For each of these species, we downloaded data from the Global Biodiversity Information Facility³⁷. We excluded data collected before 1960, which represented, on average, 41±12% of the data available, for two reasons. First, old records often lack sufficiently precise location information. Second, we wanted to avoid a potential mismatch between old observations and current climate conditions used for modelling. To complete these data and generate a dataset across the entire range of each species in Europe, we specifically performed a thorough literature review to document their occurrence from more than 600 sources’

P14 line 307 How well does the climatic data account for water availability in regions where a large proportion of plant water comes from snow fall (precipitation confounded by blowing snow, sublimation etc) and melt water from glaciers?

>> Snow is in fact an important driver of species distributions in Arctic regions, but methodological difficulties have been limiting both the quality and quantity of available snow information in species distribution models, especially for making future model projections (Niitynen & Luoto 2018). The lack of sufficiently detailed information on snow precipitation across Europe, together with the need to maintain the number of variables low to keep good model predictive power (see comment 1c of Ref3 below), prevented us from implementing this factor in our models. Its inclusion should, however, certainly be considered in future studies, which we emphasize in the revised version of the ms.

Reviewer #3 (Remarks to the Author):

Review of NCOMMS-19-39281 “Bryophytes are the canaries in the coal mine of climate warming”

In this work a process-oriented model of areas of distribution of bryophyte species is presented, in the form of a cellular automaton describing the dynamics of the distributional ranges of the species in question, as governed by niche requirements, dispersal of spores, and climate change. The work is interesting and relevant, and it is indeed one of the first examples of which I am aware in which climate change is allowed to force the parameters of a set of process-oriented distribution models. I have comments on several points, however, none of which is serious.

1. The methodology of what is in fact a relatively complicated simulation is very schematic, inconsistent or doubtful. For instance:
a. the parametrization of MigClim is inconsistent with the published literature on this package. In Line 353, is $P(\text{pixel } j)$ the same as P_{col} in eq. (1) of Engler (2012)? If so, P_{inv} in that equation (1) will be calculated from the output of BioMod? These points need to be made explicit and clear. In L 398, is n_{Disp} the same as dispSteps in Engler (2012)? In general it would be very good for the reader if authors would maintain the same symbols as in previous literature, or to explicitly state which ones have been changed. Maybe adding a table with the parameters and how they were obtained would be a good idea.

>> We fully agree and, to avoid confusion, included in the M&M section a new paragraph describing the basic MigClim methodology and its parameters, employing the same terms as in the original publication by Engler & Guisan (2009). To answer your question specifically, $P(\text{pixel } j)$ is the probability of dispersal as a function of the distance from the source, whereas P_{col} is the combined probability that a target pixel is colonized from all the potential source pixels. P_{inv} , in turn, is the probability of a cell to become colonized given its habitat suitability, which we determined in the present study from the binarized projection of the species distribution models. n_{Disp} is the total number of dispersal events, whereas DispSteps is the number of dispersal events per environmental change step. Regarding the suggestion to add a table with the parameters and their estimation, we added a new paragraph (‘Parameter estimates’) where we described how each parameter was measured or assessed.

b. Another methodological point that needs clarification is the use of so called “pseudoabsences” (L311). Which one of the at least four types (Barbet-Massin et al. 2012) of possible “pseudoabsences” is used? They are by no means equivalent, and using one or the other may change radically the interpretation of the output of a regression method, from a true probability to a relative probability, and from being able to use rigorously an AUC to better avoiding it. Be explicit about which type is being used.

>> >> Pseudo-absence points were sampled from a random selection among all points within the studied area, excluding available presence points. Following Barbet-Massin et al. (2012), 10,000 pseudo-absence points were then sampled for GLM and down-weighted to give them same overall prevalence as the presences, while for GBM, we sampled a number of pseudo-absence points identical to the number of presences. We have now made it clearer in the revised version.

c. The authors chose to model environmental tolerances using five bioclimatic variables from WorldClim. Then they used BioMod to obtain

what, strictly speaking, are response curves to climatic variables (also called ecological niches, and I assume these are used to parametrize P_{inv} in equation (1), if this is the case, please say it explicitly).

>> Yes. To make it clear, we have now added a new paragraph ('MigClim simulations') describing how MigClim was implemented, where we specified that 'For each 10 yr climatic period, pixels were identified as potential targets on the basis that they become newly climatically suitable, as inferred from the binarized SDM projections.'

Now, they use GLIM and GBM as their model algorithms. These, mostly the GBM, and with five predictors, can result in "data hugging", meaning that they will be faithful models of the observations, but not necessarily good for transferring predictions. However, this paper is very much about transferring. Would not be preferable to fit fewer variables and simpler shapes to the data points? Please discuss this matter in a couple of lines.

>> We discussed these issues in the M&M section as follows: 'A key issue with modelling responses to climate change is that we do not fully understand how models made under current conditions will transfer to future conditions. Models developed using too many predictors may run the risk of overfitting to local conditions, restricting the predictive power of the model (Peterson 2011, Werkowska et al. 2016). Tests of transferability across taxa and geographic locations have, however, failed to demonstrate consistent patterns, and a general approach to developing transferable models remains elusive (Yates et al. 2018). Here, we compared the ROC and maxTSS values computed from the test sets (20% of the data) to those observed at the level of the entire dataset, assuming that these statistics at the level of the entire dataset would substantially drop at the level of the test sets in case of severe issues of overfitting.' We reported these statistics in Table S1 and reported in the result section that no signature of overfitting could be observed, as follows: 'The climatic niche models exhibited high average True Skill Statistics (TSS) and Area Under The Curve (AUC) of a ROC plot (Receiver Operating Characteristics) statistics²⁴ of 0.78 ± 0.12 and 0.93 ± 0.05 respectively, when models were evaluated with the test sets corresponding to 20% of the data. These models did not show any apparent signature of overfitting, as a very slight increase in AUC and TSS (0.81 ± 0.13 and 0.94 ± 0.05 , respectively) was observed when these statistics were computed at the level of the entire dataset (Table S1).'

d. It is not at all clear how LDD is calculated: "Here, LDD was implemented beyond the range covered by our dispersal simulations, i.e., beyond 10km" OK, but how much is "beyond"? 0.01 km? 1,000 km? Then, in L 410 you state that you use "... two extreme values of the probability range of LDD, that is, 0 and 0.1" Are 0 and 0.1 "extreme values"? This needs explanation. It would add much clarity having a detailed description of how LDD are actually estimated.

>> We had indeed omitted to provide this information. We have now added a paragraph in the M&M section of the revised ms to clarify this as follows: 'In addition to short-distance dispersal events with a probability defined by the kernel described above, any pixel located at >10km from a potential source could be colonized by LDD. The maximum LDD distance was set to unlimited based on phylogeographic evidence 37. Following Robledo-Arnuncio et al.30, we employed the results of previous Approximate Bayesian Computation methods for LDD inference from genetic structure data in bryophytes 38,75 to define the range of LDD probability values, set to 0, 10⁻⁴, 10⁻³, 10⁻² and 10⁻¹'.

2. Finally, my personal obsession. In line 297 you call "Species distribution models" the section where you are calculating the suitability of cells (L323-324): "Finally, we computed the suitability at each pixel based on the average of the two GLM and GBM consensus models." In this section you are modeling suitability, NOT distributions. You will be modeling the actual distribution when you add dispersal, (clearly assuming interactions do not matter). What you do in the SDM section is actually to estimate one parameter (P_{inv}), the local suitability (also called the niche), for the simulation that actually estimates the distribution. Why to call the section "Species distribution models" when you are modeling climatic suitability (not "habitat suitability", as stated in fig 4. and L396, you are calculating climatic suitability)? The distinction matters. In L62-65 you state that adding dispersal adds predictive power to models. Of course it does! Simple correlative models are not SDMs, but ENMs projected to geography, and this projection models area of suitability (potential distributions). To get the actual distribution more factors are needed, like interactions and movements. However, if you fail to distinguish between a potential distribution and an actual one confusion follows. Maybe have another look at Fordham et al. (2017) to see how they distinguish between modeling distributions, and modeling niches. Now, since a large chunk of the literature does not bother with distinguishing between modeling something in geography (a distribution, with several types) and something in an abstract multidimensional space (the limits of tolerance, or "niches"), I am not going to ask that you change your terminology. I simply suggest you consider the possibility.

>> We thanks the reviewer for his comment, which indeed is pertinent. Although we disagree that there is a clear terminological rule in the literature around the use of the terms ENM versus SDM, we agree that in our study it is important to distinguish the steps of calculating habitat suitability (mainly climatic) and of then adding dispersal constraints. Whether calling these ENM and then only SDM when dispersal is added is still subject to debate, and we appreciate the flexibility of the reviewer in this regard, yet we agree that calling the first step 'climatic suitability' makes more sense in our case, as it says more precisely what is actually done. We have therefore revised the ms accordingly, and followed your suggestion to rename the modelling section, which we now entitled 'Deriving climatic suitability maps'. We have accordingly replaced 'species distribution models' by 'climatic suitability models' throughout the ms, avoided using any related acronym, and acknowledged that in the literature, the different terms are very often (though not always) used equivalently for the same models (see also Guisan et al. 2017, Araujo et al. 2019).

3. The paper does not cite several quite relevant references. Maybe the journal imposes a limit to the number of references, but I suggest that the authors consider citing the papers I will suggest below.

>> Thank you for the useful references, which we are happy to include.

4. Minor points.

a. L 81. What are "intraspecific variations in wind conditions"?

>> 'intraspecific' was superfluous and we reworded the sentence as follows: 'Implementing a hybrid statistical-mechanistic approach accounting for temporal and spatial variation of both climatic conditions and wind connectivity to predict bryophyte distributions under climate change across Western Europe, we show that rates of range loss largely exceed the proportion of newly suitable habitats that could effectively be colonized, suggesting that even highly dispersive organisms such as bryophytes are not equipped to track the rates of ongoing climate change in the course of the next decades'.

b. L. 189-190, you state that your approach "... over-estimate colonization rates because all climatically suitable pixels for the present time served as starting points in dispersal simulations," Well, in the first place, you are surely not talking about overestimating "rates" but overestimating the result of the simulation. In the second place, if you are modeling the future distribution of a species, you start your simulation from where it exists today. I think what you did is right, starting the spread from every recorded locality, since you are asking questions about future distributions considering movements. The occupied localities are the initial conditions for the cellular automaton.

>> **We agree and rephrased our sentence as follows: 'These limitations include, most importantly, the assumptions that dispersal is isotropic, that newly colonized cells are readily considered as sources, thereby ignoring demography, that there is no competition and that microclimatically suitable pixels serve as migration sources'.**

c. L. 212, "...under the most optimistic dispersal conditions" What are "most optimistic"? One needs to see the parameters...

>> **We meant here that these results are obtained with the dispersal kernels with the highest probabilities of dispersal (in terms of height of release, wind speed and probability of LDD), but this is actually redundant with the comment that our approach tends to over-rather than under-estimate the number of colonization events, and we therefore deleted 'under the most optimistic dispersal conditions'.**

d. L 204-205. You state that wider distribution ranges imply wider climatic niches. Not necessarily. Imagine a species widely distributed over northern Russia, with a very narrow climatic niche. Or the opposite, a species with a wide range of climatic tolerances, endemic to a single island. This statement is not convincing.

>> **We agree and rephrased this part completely. It now reads: 'We suggest that this difference is due to the much wider distribution range, higher dispersal capacities and, potentially, broader climatic niche of Mediterranean bryophytes as compared to their angiosperm counterpart. This is best illustrated by the large differences in rates of local endemism between the two groups, as more than 60% of Mediterranean endemic angiosperm species are restricted to a single region³⁶ and are, hence, prone to extinction if they fail to colonize newly suitable areas, whereas there is no local endemism reported to date in the Mediterranean bryophyte flora³⁷'.**

e. L. 392, by "integrated" you mean summed over pixels?

>> **Yes—we rephrased the sentence accordingly ('Finally, $P(\text{pixel } j)$ values are computed at increasing distances from the source and combined to generate a global probability of colonization P_{col} from n potential source pixels').**

Bibliography suggested

Colwell, R. K., and T. F. Rangel. 2010. A stochastic, evolutionary model for range shifts and richness on tropical elevational gradients under Quaternary glacial cycles. *Philosophical Transactions of the Royal Society B: Biological Sciences* 365:3695-3707.

Hooten, M. B., and C. Wikle. 2008. A hierarchical Bayesian non-linear spatio-temporal model for the spread of invasive species with application to the Eurasian Collared-Dove. *Environmental and Ecological Statistics* 15:59-70.

Ingenloff, K., C. M. Hensz, T. Anamza, V. Barve, L. P. Campbell, J. C. Cooper, E. Komp et al. 2017. Predictable invasion dynamics in North American populations of the Eurasian collared dove *Streptopelia decaocto*. *Proceedings of the Royal Society B: Biological Sciences* 284:20171157.

Qiao, H., E. E. Saupé, J. Soberón, A. T. Peterson, C. E. Myers, D. C. Collar, and J. L. Bronstein. 2016. Impacts of Niche Breadth and Dispersal Ability on Macroevolutionary Patterns. *The American Naturalist* 188:149-162.

Schurr, F. M., J. Pagel, J. S. Cabral, J. Groeneveld, O. Bykova, R. B. O'Hara, F. Hartig et al. 2012. How to understand species' niches and range dynamics: a demographic research agenda for biogeography. *Journal of Biogeography* 39:2146-2162.

Reviewers' Comments:

Reviewer #1:

Remarks to the Author:

The paper has been revised in response to my comments and those of 2 other reviewers on an earlier version. The authors have done a good job with the responses and the revisions. They have justified the novelty of the approach and dealt well with the methodological, conceptual and presentational issue I mentioned. I think it is now a great contribution to the field.

I have one, very minor, thought. The title is in the present tense and so suggests the paper is reporting current tracking of climate change. It might be good to put it into the future perfect tense: Even bryophytes will lag behind climate change

Reviewer #2:

Remarks to the Author:

The authors seem to have made a good job of addressing the comments made in the various reviews in a thoughtful and considered manner. I feel that the manuscript is much improved and suitable for publication.

I appreciate that microclimatic scaling is currently very difficult to resolve but I think it is important to acknowledge the issue so as not to mislead readers, especially when it comes to tiny plants like bryophytes. The authors have done this. They have also satisfactorily addressed the other comments I made.

I have two minor comments, the first is a suggested rephrasing

Line 44

While climate change is making some current habitats unsuitable, it is also expected to create newly suitable areas for species to occupy.

The other two relate to very recent publications which the authors might find relevant and want to reference.

A new phylogenetic study (Biersma et al 2020) which details the importance of prevailing wind patterns on bryophyte dispersal in a cosmopolitan moss species.

And a paper by Perera-Castro et al (2020), which shows that Antarctic bryophytes have quite high temperature optima which I think is relevant to the comments on line 79 "Furthermore, bryophyte species of temperate biomes exhibit lower optima and tolerance to warm temperatures than their angiosperm counterparts" (but see Perrera et al 2020)

Biersma et al 2020 Latitudinal Biogeographic Structuring in the Globally Distributed Moss *Ceratodon purpureus* Front. Plant Sci. doi: 10.3389/fpls.2020.502359

Perera-Castro et al 2020 It Is Hot in the Sun: Antarctic Mosses Have High Temperature Optima for Photosynthesis Despite Cold Climate. Front. Plant Sci. doi: 10.3389/fpls.2020.01178

Sharon Robinson

Reviewer #3:

Remarks to the Author:

I reviewed your response to my comments and found them satisfactory.

REVIEWERS' COMMENTS

Reviewer #1 (Remarks to the Author):

The paper has been revised in response to my comments and those of 2 other reviewers on an earlier version. The authors have done a good job with the responses and the revisions. They have justified the novelty of the approach and dealt well with the methodological, conceptual and presentational issue I mentioned. I think it is now a great contribution to the field.

I have one, very minor, thought. The title is in the present tense and so suggests the paper is reporting current tracking of climate change. It might be good to put it into the future perfect tense: Even bryophytes will lag behind climate change

>> The title was changed as follows: 'Bryophytes are predicted to lag behind future climate change despite their high dispersal capacities' following the editor's suggestion.

Reviewer #2 (Remarks to the Author):

The authors seem to have made a good job of addressing the comments made in the various reviews in a thoughtful and considered manner. I feel that the manuscript is much improved and suitable for publication.

I appreciate that microclimatic scaling is currently very difficult to resolve but I think it is important to acknowledge the issue so as not to mislead readers, especially when it comes to tiny plants like bryophytes. The authors have done this. They have also satisfactorily addressed the other comments I made.

I have two minor comments, the first is a suggested rephrasing

Line 44

While climate change is making some current habitats unsuitable, it is also expected to create newly suitable areas for species to occupy.

>> Thanks for rewording the sentence, which was modified as suggested.

The other two relate to very recent publications which the authors might find relevant and want to reference.

A new phylogenetic study (Biersma et al 2020) which details the importance of prevailing wind patterns on bryophyte dispersal in a cosmopolitan moss species.

And a paper by Perera-Castro et al (2020), which shows that Antarctic bryophytes have quite high temperature optima which I think is relevant to the comments on line 79 "Furthermore, bryophyte species of temperate biomes exhibit lower optima and tolerance to warm temperatures than their angiosperm counterparts" (but see Perrera et al 2020)

>> Thanks for the references. The paper by Perera-Castro et al. is intriguing and at odds with previous reports on comparatively low temperature optima in bryophytes. This definitely calls for further research on this topic and we changed the relevant sentence of the introduction as you suggested. Since we already had 80 references, we did not cite the Biersma et al. paper, which focuses on the phylogeography of the cosmopolitan moss *Ceratodon purpureus* and wherein evidence that patterns of genetic variation have been shaped by wind current is missing.

Biersma et al 2020 Latitudinal Biogeographic Structuring in the Globally Distributed Moss *Ceratodon purpureus* Front. Plant Sci. doi: 10.3389/fpls.2020.502359

Perera-Castro et al 2020 It Is Hot in the Sun: Antarctic Mosses Have High Temperature Optima for Photosynthesis Despite Cold Climate. Front. Plant Sci. doi: 10.3389/fpls.2020.01178

Sharon Robinson

Reviewer #3 (Remarks to the Author):

I reviewed your response to my comments and found them satisfactory.